# Effect of Activator and Outgoing Ligand Nature on the Catalytic Behavior of Bis(phenoxy-imine) Ti(IV) Complexes in the Polymerization of Ethylene and Its Copolymerization with Higher Olefins

**DOI:** 10.3390/polym14204397

**Published:** 2022-10-18

**Authors:** Svetlana Ch. Gagieva, Kasim F. Magomedov, Vladislav A. Tuskaev, Vyacheslav S. Bogdanov, Dmitrii A. Kurmaev, Evgenii K. Golubev, Gleb L. Denisov, Galina G. Nikiforova, Maria D. Evseeva, Daniele Saracheno, Mikhail I. Buzin, Pavel B. Dzhevakov, Viktor I. Privalov, Boris M. Bulychev

**Affiliations:** 1Department of Chemistry, M. V. Lomonosov Moscow State University, 1 Leninskie Gory, 119992 Moscow, Russia; 2A. N. Nesmeyanov Institute of Organoelement Compounds, Russian Academy of Sciences, Vavilova Str., 28, 119991 Moscow, Russia; 3Enikolopov Institute of Synthetic Polymer Materials, Russian Academy of Sciences, Profsoyuznaya Str., 70, 117393 Moscow, Russia; 4Kurnakov Institute of General and Inorganic Chemistry, Russian Academy of Sciences, 31, Leninsky Prospect, 119991 Moscow, Russia

**Keywords:** FI catalysts, titanium(IV), zirconium(IV), polymerization, copolymerization, ultra-high molecular weight polyethylene, ethylene/higher olefin copolymer

## Abstract

A series of bis(phenoxy-imine) (FI) titanium(IV) and zirconium(IV) complexes have been synthesized. The effect of the nature of the activator (MAO, combinations Et_n_AlCl_3-n_ + Bu_2_Mg and iBu_3_Al + [Ph_3_C]^+^[B(C_6_F_5_)_4_]^−^) on the catalytic activity and properties of the resulting polymers was studied. It was found that Ti-Fi complexes, despite the nature of the outgoing ligands (Cl or iPrO) in the presence of Al/Mg activators, effectively catalyze the polymerization of ethylene (with the formation of UHMWPE); copolymerization of ethylene with 1-octene (with the formation of ultra-high molecular weight copolymers); and the ternary copolymerization of ethylene, propylene and 5-vinyl-2-norbornene (with the formation of polyolefin elastomers). It has been shown that Zr-FI complexes are not activated by these Al/Mg compositions. The resulting UHMWPE can be processed by a solventless method into high-strength and high-modulus oriented films; however, their mechanical characteristics do not exceed those obtained using MAO.

## 1. Introduction

Phenoxyimine complexes of Group 4 metals (FI-catalysts) are obviously the most well-studied class of post-metallocene catalysts for olefin polymerization, comparable in productivity to the most “successful” metallocenes [1,2,3,4]. Catalytic systems based on FI complexes effectively catalyze the polymerization of all basic olefin monomers as well as the processes of their copolymerization, with high activity and unprecedented selectivity. Currently, technologies for the industrial production of polymer materials using FI catalysts have been developed [5]. It is important that these catalytic systems are capable of producing ultra-high molecular weight polyethylene including polymers with a low degree of entanglement of macromolecules, which opens up opportunities for their processing by the solventless solid-phase method into high-strength high-modulus materials [6,7,8,9,10].

It is postulated that during the activation neutral FI complexes are transformed into coordinatively unsaturated, highly electrophilic monoalkyl cationic complexes, which mediate olefin polymerizations via a coordination-insertion mechanism. Changing activation methods is one of the main ways to control the properties of the resulting polymers (the molecular weight, molecular weight distribution, degree of entanglement, content and distribution of comonomer units in the polymer chain). The authors of [11], comparing the effect of various alkylalumoxanes (MAO, polymethylaluminoxane-improved performance (PMAO), modified methylaluminoxane type 12 (MMAO12) and type 3A (MMAO3A)) on solid-state processability (uniaxial solid-state deformation) of UHMWPE samples, came to the conclusion that the best mechanical characteristics are achieved when using MMAO-12.

Alkylaluminoxane derivatives are traditional activators for FI catalysts. However, the use of expensive MAO derivatives in significant excess (as a minimum, 500–1000 eq. per mole of transition metal) has been a critical concern in commercial operation. Therefore, in recent years, significant efforts of researchers have been directed to the development of new types of activators. In particular, attempts have been made (generally not very successful) to use traditional trialkylaluminum and alkylaluminum chlorides for FI catalysts activation [12,13,14,15]. Many efforts have been made to replace MAO with non-coordinating anions (e.g., [B(C_6_F_5_)_4_]^−^); however, their cost is also very high. When FI catalysts are activated by iBu_3_Al and Ph_3_CB(C_6_F_5_)_4_, the imine moiety is reduced to an amine by iBu_3_Al (or contaminant iBu_2_AlH) accompanied by isobutene formation (FI catalysts and iBu_3_Al are mixed for 10 min before polymerization) [16,17], resulting in phenoxy–amine complexes. The reduced species exhibited a number of interesting polymerization characteristics owing to their unusual N donors, iBu_2_Al–N, which will be bulkier and weaker as a coordinating donor than the imine-N [18]. Zirconium(IV)dichloride complexes having two phenoxy−imine chelate ligands polymerized ethylene with activity up to 550 kg/mmol of cat·h using MAO as a cocatalyst; PE have a viscosity average molecular weight (Mv) value of 0.9 × 10^4^. The same pre-catalysts, activated with Ph_3_CB(C_6_F_5_)_4_/i-Bu_3_Al, produce extremely high molecular weight polyethylene, Mv 5.05⸱10^6^ while the activity remained quite high 11 kg/mmol of cat·h [19].

Some progress has been made in the development of activators containing neither alumoxanes nor perfluorophenylborates. MgCl_2_/AlR_n_(OEt)_3-n_ or MgCl_2_/AlR_n_Cl(OEt)_2-n,_, obtained by reaction of AlR_3_ or Et_2_AlCl with adducts of MgCl_2_ and ethanol, have been shown to be effective for the immobilization and activation of metallocenes and FI catalysts without the use of methylaluminoxane or a borate activator [1]. Ti-, Zr-, and V-based FI catalysts combined MgCl_2_/R’_n_Al(OR)_3-n_ formed highly active supported single-site catalysts demonstrated superior catalytic properties, compared to the corresponding homogeneous methylaluminoxane-(Ti and Zr-FI catalysts) or alkylaluminum-activation systems (V-FI catalysts) [20].

In our early works, we demonstrated the possibility of using mixtures of alkylaluminium chlorides with dibutylmagnesium for the activation of phenoxy-imine pre-catalysts [21,22]. In such catalytic systems, MgCl_2_ is formed in situ [23,24], however, in much smaller amounts. The optimal molar ratio Ti/Al/Mg is 1/100/300.

In the vast majority of cases, phenoxy-imine pre-catalysts are dichloride complexes of Group 4 metals, and the role of the “leaving” σ-ligand other than chlorine or hydrocarbyl has been little studied. In one of the few works [25], it was shown that the Zr-amido complex **FI_2_Zr(NMe_2_)** activated with Me_3_Al/Ph_3_C^+^B(C_6_F_5_)_4_^−^ in the copolymerization of ethylene and octene-1 surpasses the benzyl and chloride analogs in terms of activity and degree of inclusion of the comonomer.

Titanium dialkoxide phenoxy-imine complexes, according to the literature data, exhibit moderate activity in the presence of alkylalumoxanes [26,27,28]; however, it is inferior to dichloride complexes by orders of magnitude. One of the goals of this work is to study the catalytic activity of alkoxide analogs of dichloride phenoxy-imine complexes activated with MAO, mixtures {Et_n_AlCl_3-n_ + Bu_2_Mg} and perfluorophenylborates.

The main aim of present work is to compare catalytic performance of Ti(IV) and Zr(IV) phenoxy- imine complexes in the ethylene polymerization and its copolymerization with 1-octene and ternary copolymerization of ethylene/propylene/5-vinyl-2-norbornene as well as the properties of the resulting polymers, depending on the nature of the activator (MMAO-12, combinations Et_n_AlCl_3-n_+Bu_2_Mg and iBu_3_Al + [Ph_3_C]^+^[B(C_6_F_5_)_4_]).

## 2. Experiments

All manipulations with air-sensitive materials were performed using standard Schlenk techniques. Argon and ethylene of special-purity grade (Linde gas) were dried by purging through a Super Clean™ Gas Filters.

Toluene was distilled over Na/benzophenone ketyl. Diethylaluminum chloride, ethylaluminum sesquichloride, triisobutylaluminum and di-*n*-butylmagnesium (Aldrich) were used as 1.0 M solution in heptane. Trimethylaluminum (Aldrich) was used as 2.0 M solution in hexanes. Methylaluminoxane (Sigma-Aldrich) was used as 10 wt% solutions in toluene. The preparation of the ligands **L1** and **L2** followed the procedure described in [29,30]; their properties corresponded to literature data.

NMR spectra were recorded on Bruker AMX-400 instrument. Elemental analysis (C, H, Cl) was performed by the microanalytical laboratory at A. N. Nesmeyanov Institute of Organoelement Compounds on Carlo Erba-1106 and Carlo Erba-1108 instruments. The contents of Ti were performed by X-ray fluorescence analysis on the VRA-30 device (Karl Zeiss, Germany).

### 2.1. Bis[N-(3,5-di-tert-butylsalicylidene)-anilinato]titanium(IV) Dichloride, (L1)_2_TiCl_2_

In a 50 mL flame-dried Schlenk flask, ligand **L1** (0.309 g, 0.10 mmol) was dissolved in anhydrous toluene (10 mL). A solution of TiCl_2_(OiPr)_2_ (0.117 g, 0.05 mmol) in toluene (10 mL) was added to the resulting solution under stirring in an argon atmosphere. The solution was stirred for 14 h at room temperature, the resulting red precipitate was filtered off, washed with hexane (5 mL) and dried in vacuum. The yield of the complex was 0.52 g (71%). Calculated (%) for C_42_H_52_Cl_2_N_2_O_2_Ti (734.29): C, 68.57; H, 7.12; N, 3.81; Cl, 9.64; Ti 6.51. Found (%): C, 68.53; H, 7.09; N, 3.77; Cl 9.58; Ti, 6.46. ^1^ H NMR (toluene-d8), δ (ppm): 1.38 (s, 18H, t-Bu), 1.68 (s, 18H, t-Bu), 7.15–6.95 (m, 12H, Ar), 7.64 (s, 2H, Ar), 8.12 (s, 2H, N=CH). ^13^C (toluene-d8), δ (ppm): 19.46, 19.65, 19.85, 25.86, 29.52, 29.63, 29.93, 31.22, 31.27, 33.81, 39.06, 78.17, 123.13, 124.50, 124.74, 127.34, 127.57, 127.58, 127.81, 127.82, 128.00, 129.51, 137.10, 167.70. IR, ν, cm^–1^: 1642 (C=N); 529 (Ti—O); 447 (Ti—N).

Obtained by a similar method:

### 2.2. Bis[N-(3,5-di-tert-butylsalicylidene)-(2,3,5,6-tetrafluoroanilinato)]titanium(IV) Dichloride, (L2)_2_TiCl_2_

The yield—0.75 g (85%). Calculated (%) for C_42_H_44_Cl_2_F_8_N_2_O_2_Ti (878.21): C, 57.35; H, 5.04; N, 3.18; Cl 8.06; F 17.28; Ti, 5.44. Found (%): C, 57.28; H, 4.98; N, 3.15; Cl 8.01; Ti, 5.39. ^1^ H NMR (CDCl_3_), δ (ppm): 1.33 (s, 9H, t-Bu), 1.48 (s, 9H, t-Bu), 6.92 (tt, 3JHF = 10 Hz, 4JHF = 7 Hz, 1H, p-C_6_F_4_), 7.22 (d, 4JHH = 2.3 Hz, 1H, Ar), 7.55 (s, 1H, Ar), 8.83 (s, 1H, N=CH), 12.82 (s, 1 H). ^13^C (CDCl_3_), δ (ppm): 29.27, 29.39, 39.29, 31.39, 34.22, 35.18, 101.91, 102.14, 102.36, 117.85, 127.65, 129.98, 137.58, 141.12, 158,89, 171,68. IR, ν, cm^–1^: 1610 (C=N); 520 (Ti—O); 450 (Ti—N).

### 2.3. Bis[N-(3,5-di-tert-butylsalicylidene)-anilinato]zirconium(IV) Dichloride (L1)_2_ZrCl_2_

The yield of the complex was 0.37 g (45%). The single crystal for X-rays was obtained by recrystallization of the resulting crystalline powder in a 3 mL hexane. Calculated (%) for C_42_H_52_Cl_2_N_2_O_2_Zr (776.24): C, 64.76; H, 6.73; N, 3.60; Cl, 9.10; Zr, 11.71. Found (%): C, 64.72; H, 6.67; N, 3.55; Cl 9.02; Ti, 5.39. ^1^H NMR (toluene-d8), δ (ppm): 1.36 (s, 18H, t-Bu), 1.67 (s, 18H, t-Bu), 7.07–6.74 (m, 12H, Ar), 7.62 (s, 2H, N=CH). ^13^C NMR (101 MHz, toluene-d8) δ 169.61, 161.02, 153.10, 138.79, 138.08, 130.02, 129.05, 128.24, 128.04, 127.80, 127.56, 125.45, 123.37, 123.12, 121.96, 71.81, 70.54, 35.30, 33.88, 31.45, 29.84, 29.57, 27.34, 22.80, 14.11.

### 2.4. Bis[N-(3,5-di-tert-butylsalicylidene)-(2,3,5,6-tetrafluoroanilinato)] Zirconium (IV) Dichloride, (L2)_2_ZrCl_2_

The yield of the complex was 0.45 g (54%). Calculated (%) for C_42_H_44_Cl_2_F_8_N_2_O_2_Zr (920.17): C, 54.66; H, 4.81; N, 3.04; Cl, 7.68; F, 16.47; Zr, 9.88. Found (%): C, 54.62; H, 4.76; N, 2.98; Cl, 7.64; Zr, 9.85. ^1^ H NMR (400 MHz, toluene-d8), δ (ppm): δ 1.30 (s, 9H), 1.60 (s, 9H), 6.14 (s, 1H), 7.01 (d, J = 2.5 Hz, 2H), 7.65 (s, 1H). ^13^C NMR (101 MHz, toluene-d8), δ, (ppm) 171.79, 159.34, 140.97, 137.76, 129.99, 118.18, 101.70, 35.23, 34.02, 31.24, 31.11, 29.39.

### 2.5. Bis[N-(3,5-di-tert-butylsalicylidene)-anilinato]titanium(IV) Diisopropoxide, (L1)_2_Ti(OiPr)_2_

In a 50 mL flame-dried Schlenk flask, ligand **L1** (0. 309 g, 0.10 mmol) was dissolved in anhydrous toluene (10 mL). A solution of Ti(OiPr)_4_ (0.142 g, 0.05 mmol) in toluene (10 mL) was added to the resulting solution under stirring in an argon atmosphere. The solution was stirred for 14 h at room temperature, the resulting orange solution was evaporated to dryness, the residue was recrystallized from hexane. The precipitated complex was filtered off, washed with hexane (5 mL) and dried in a vacuum. The yield of the complex was 0.63 g (68%). The single crystal for X-rays was obtained by recrystallization of the resulting crystalline powder in a mixture 5 mL hexane and 1 mL toluene. Calculated (%) for C_48_H_66_N_2_O_4_Ti (782.45): C, 73.64; H, 8.50; N, 3.58; Ti, 6.11. Found (%): C, 73,61; H, 8,45; N, 3,52; Ti, 6,06. ^1^H NMR (400 MHz, CDCl_3_) δ (ppm); 1.31 (s, 9H, t-Bu), 1.37 (s, 9H, t-Bu), 1.54 (s, 12H, CH_3_), 4.05 (1H, CH), 4.76 (1H, CH), 7.45–7.02 (m, 12H, Ar), 8.68 (s, 2H, N=CH). ^13^C NMR (101 MHz, CDCl_3_), δ (ppm): 168.26, 167.48, 166.31, 163.84, 162.02, 160.62, 158.27, 155.21, 154.45, 153.87, 148.75, 140.57, 140.02, 138.70, 138.21, 138.02, 137.89, 137.43, 136.99, 129.92, 129.44, 129.41, 129.35, 129.06, 128.42, 128.35, 128.25, 128.11, 128.03, 127.90, 127.80, 127.63, 126.83, 126.55, 125.54, 125.32, 125.07, 125.01, 123.67, 123.13, 123.07, 121.62, 121.28, 121.20, 118.31, 35.30, 35.13, 35.08, 34.97, 34.21, 34.17, 34.04, 33.97, 31.54, 31.50, 31.47, 29.81, 29.56, 29.51, 29.47, 29.45, 26.57, 25.88, 24.76, 21.49. IR, ν, cm^–1^: 1595 (C=N); 512 (Ti—O).

Obtained by a similar method:

### 2.6. Bis[N-(3,5-di-tert-butylsalicylidene)-(2,3,5,6-tetrafluoroanilinato)]titanium(IV) Diisopropoxide, (L2)_2_Ti(OiPr)_2_

The yield of the complex was 0.65 g (71%). Calculated (%) for C_48_H_58_F_8_N_2_O_4_Ti (926.37): C, 62.20; H, 6.31; N, 3.02; F, 16.40; Ti, 5.16. Found (%): C, 62.13; H, 6.28; N, 2.96; Ti, 5.13. ^1^ H NMR (CDCl_3_), δ (ppm): 1.26 (s, 9H, t-Bu), 1.32 (s, 9H, t-Bu), 1.50 (s, 12H, CH_3_), 4.81 (2H, CH), 6.93 (tt, 3JHF = 10 Hz, 4JHF = 7 Hz, 1H, p-C_6_F_4_), 7.60 (s, 1H, Ar), 7.15 (s, 1H, Ar), 8.31 (s, 1H, N=CH). ^13^C NMR (CDCl_3_), δ (ppm): 172.81, 171.72, 162.15, 158.89, 141.11,140.07, 138.78, 137.56, 131.25, 131.19, 129.97, 128.74, 128.67, 120.89, 120.20, 117.84, 102.14, 101.94, 191.64, 76.22, 35.33, 35.18, 35.07, 34.22, 34.15, 31.33, 31.35, 31.24, 29.45, 29.38, 26.55, 25.79. IR, ν, cm^–1^: 1590 (C=N); 530 (Ti-O).

### 2.7. Bis[N-(3,5-di-tert-butylsalicylidene)-(2,3,5,6-tetrafluoroanilinato)]zirconium(IV) Diisopropoxide, (L1)_2_Zr(OiPr)_2_

The yield of the complex was 0.49 g (56%). The single crystal for X-rays was obtained by recrystallization of the resulting crystalline powder in a mixture 2 mL hexane and 1 mL toluene. Calculated (%) for C_48_H_66_N_2_O_4_Zr (824.41): C, 69.77; H, 8.05; N, 3.39; Zr, 11.04. Found (%): C, 69.72; H, 8.01; N, 3.35; Zr, 11.00. ^1^H NMR (400 MHz, C_6_D_6_), δ (ppm): 1.42 (s, 9H, t-Bu), 1.52 (s, 9H, t-Bu), 1.74 (s, 12H, CH_3_), 2.02 (m, 2H, CH), 6.91–7.70 (m, 12H, Ar), 7.88 (s, 2H, N=CH). ^13^C NMR (101 MHz, C_6_D_6_), δ (ppm): 169.61, 161.02, 153.10, 138.79, 138.08, 130.02, 129.05, 128.24, 128.04, 127.80, 127.56, 125.45, 123.37, 123.12, 121.96, 71.81, 70.54, 35.30, 33.88, 31.45, 29.84, 29.57, 27.34.

### 2.8. Bis[N-(3,5-di-tert-butylsalicylidene)-(2,3,5,6-tetrafluoroanilinato)] Zirconium (IV) Diisopropoxide, (L2)_2_Zr(OiPr)_2_

The yield of the complex was 0.36 g (38%). The single crystal for X-rays was obtained by recrystallization of the resulting crystalline powder in 2 mL hexane. Calculated (%) for C_48_H_58_F_8_N_2_O_4_Zr (968.33): C, 59.42; H, 6.03; N, 2.89; F, 15.67; Zr, 9.40. Found (%): C, 59.35; H, 5.93; N, 2.82; Zr, 9.38. IR, ν, cm^–1^: 1610 (C=N); 540 (Ti—O). 1H NMR (400 MHz, toluene-d8), δ, (ppm): 1.22 (s, 3H), 1.24 (s, 3H), 1.31 (s, 3H), 1.32 (s, 3H), 1.36 (s, 18H), 1.48 (s, 18H), 4.58 (s, 2H), 6.10 (s, 2H), 6.91 (s, 2H), 7.57 (s, 2H), 7.62 (s, 2H). ^13^C NMR (101 MHz, toluene-d8), δ (ppm): 175.18, 161.97, 139.20, 131.90, 129.58, 121.44, 101.83, 72.53, 35.29, 33.95, 31.16, 29.49, 26.99.

### 2.9. X-ray Crystal Structure Determination

X-ray diffraction data for **(L1)_2_Ti(OiPr)_2_**, **(L1)_2_Zr(OiPr)_2_**, **(L2)_2_Zr(OiPr)_2_**, and **(L1)_2_Zr(Cl)_2_** were collected at 100 K with a Bruker D8 Quest CMOS diffractometer using the graphite monochromated Mo-Kα radiation (λ = 0.71073 Å, ω-scans). Using Olex2 [31], the structures were solved with the ShelXT [32] structure solution program using Intrinsic Phasing and refined with the XL refinement package [33] using Least Squares minimization against *F*^2^ in anisotropic approximation for non-hydrogen atoms. Hydrogen atoms were calculated and refined in the isotropic approximation within the riding model. Crystal data and structure refinement parameters are given in Table S1. CCDC 2189402–2189405 contain the supplementary crystallographic data for of **(L1)_2_Ti(OiPr)_2_**, **(L1)_2_Zr(OiPr)_2_**, **(L2)_2_Zr(OiPr)_2_**, and **(L1)_2_Zr(Cl)_2_**, respectively.

### 2.10. Polymerization Experiments

#### 2.10.1. Polymerization with MAO or {Et_n_AlCl_3-n_ + Bu_2_Mg}

The ethylene polymerization was performed in a 450-mL reactor (Parr Instrument Co., Moline, IL, USA) equipped with a magnetic stirrer and inlets for loading components of catalytic systems and ethylene at a total ethylene and toluene vapors pressure of 1.7 atm. The ethylene polymerization technique is described in detail in [34].

#### 2.10.2. Polymerization with (^i^Bu)_3_Al/CPh_3_^+^B(C_6_F_5_)_4_^−^

Ninety mL of anhydrous toluene was introduced at room temperature into the pre-prepared reactor. In a Schlenk tube, a solution of pre-catalyst (5 · 10^−6^ M) in 5 mL of anhydrous toluene and a 1M solution of iBu_3_Al (0.04 mL) were mixed and stirred for 30 min at room temperature. Then, a solution of Ph_3_CB(C_6_F_5_)_4_ (9.23 mg, 1 · 10^−5^ M) in toluene (5 mL) was added and stirred for 1 min, and the resulting mixture was added into the reactor to initiate polymerization. All other manipulations were carried out by analogy with the procedure described in previous section.

#### 2.10.3. Copolymerization of Ethylene and 1-Octene

One hundred mL of toluene and ten mL of 1-octene were introduced at room temperature into the pre-prepared reactor under stirring; the mixture was saturated with ethylene; next, a required amount of a cocatalyst {1.5Et_3_Al_2_Cl_3_ + Bu_2_Mg}—was introduced; the resulting mixture was stirred for 5 min. Polymerization was initiated by adding a precatalyst solution in 1 mL of toluene to the reaction mixture. All other manipulations were carried out by analogy with the procedure described in section “Polymerization with MAO or {Et_n_AlCl_3-n_ + Bu_2_Mg}”.

Copolymerization of ethylene, propylene, and 5-vinyl-2-norbornene is described in detail in [34].

### 2.11. Polymer Evaluation Methods

DSC was performed by a differential scanning calorimeter DSC-822e (Mettler-Toledo, Switzerland) at a heating rate 10 °C/min in air or in argon. The heating cycle was performed twice.

Viscosity-average molecular weight of synthesized UHMWPE samples was calculated with the Mark-Houwink equation [35].

The mechanical characteristics of the oriented materials prepared from the synthesized polymers were evaluated using the oriented tapes obtained by a solid-state processing of UHMWPE nascent reactor powders. The monolithic tapes uniform over the entire length (100 mm in thickness and 10 mm in width) were formed at a pressure and shear deformation below the polymer melting point (124–126 °C). The tapes were subjected to uniaxial drawing while using a Spinline Daca equipment. The drawing temperature was set 4 °C below the polymer melting point. The mechanical characteristics of the tapes were measured with a Hounsfield H1KS machine at the gauge length of the tested samples 120 mm with 2 mm/min initial deformation rate. The reported values were the average of at least 8 samples.

Particle size analysis of UHMWPE nascent reactor powders was carried out with laser diffraction particle size analyzer Mastersizer 3000 (Malvern Instruments, Malvern, UK). The sample dispersion in acetone was prepared in the Hydro EV device at room temperature and 1100 rpm.

Scanning electron microscopy investigations of morphologies of nascent reactor powders were carried out with a high-resolution Tescan VEGA3 SEM operated at 5 kV. As-polymerized particles were carefully deposited on SEM stubs, and the samples were coated with gold by a sputtering technique. GPC analysis of the polymer samples was carried out with a Waters GPCV-2000 chromatograph equipped with two columns (PL-gel, 5 m J Mixed-C, 300 °C 7.5 mm) and a refractometer.

## 3. Results and Discussion

Phenoxyimine ligands **L1** and **L2** containing bulky *tert*-butyl groups in the *ortho*- and *para*-positions to the phenolic hydroxyl were used as objects of study in this work. These substituents protect the metal center, stabilize the catalytically active sites, and ensure good solubility of the complex. The imine fragment is represented by the residue of aniline and 2,3,5,6-tetrafluoroaniline. The latter will allow us to find out whether the “fluorine effect”-obtaining a polymer with a monomodal distribution under conditions of “living” polymerization-manifests itself when MAO is replaced by binary Al/Mg activators.

The ligands **L1** and **L2** were obtained by acid-catalyzed condensation of 3,5-di-*tert*-butylsalicylic aldehyde with the corresponding anilines (Figure 1), their properties correspond to the literature data [29,30], and the spectra are given in Supplementary Information (Appendix A).

Titanium (IV) dichloride complexes were obtained by interaction of ligands with TiCl_2_(OiPr)_2_ in anhydrous toluene. Zirconium (IV) dichloride complexes were synthesized by the reaction of ZrCl_4_ with two equivalents of sodium salt of the corresponding phenoxy-imine ligands.

The desired alkoxide complexes were obtained by the interaction of ligands with titanium or zirconium tetraisopropoxides in anhydrous toluene by elimination of isopropyl alcohol.

The composition and structure of the synthesized complexes were confirmed by elemental analysis, IR and NMR spectroscopy. In the IR spectra of the complexes, there are no absorption bands of hydroxyl groups, which indicates the formation of a Ti-O bond. The intense peak around 575 cm^−1^ can be attributed to the stretching vibrations of the Ti bond with the oxygen atoms of the ligand [36]. The downfield shift of the ν_(C=N)_ band compared to the ligand as well as the appearance of a new band in the 470–488 cm^−1^ region indicate the formation of the Ti-N coordination bond [37].

In the ^1^H NMR spectra of the complexes, the signals of hydroxyl groups also disappear. The proton signals of the imine fragment (CH=N) are shifted downfield by 0.3–1.1 ppm, which also confirms the complex formation. In the spectra of dialkoxide complexes, in addition to the signals of the ligand protons, there appears a set of signals assigned to the protons of the iso-propoxy groups.

Thermal gravimetric analysis (TGA) was undertaken on complex **(L2)_2_TiCl_2_** and corresponding ligand (Figure 1). The results indicate a significant thermal stability of the complex at least up to a temperature of 100 °C. It is of interest to note that ligand **L2** is thermally less stable as compared to complex **(L2)_2_TiCl_2_**.

For the complexes **(L1)_2_Ti(OiPr)_2_ (A)**, **(L1)_2_Zr(OiPr)_2_ (B)**, **(L2)_2_Zr(OiPr)_2_ (C)**, and **(L1)_2_Zr(Cl)_2_ (D)**, we managed to grow single crystals suitable for X-ray diffraction studies; the results are shown in Figure 2. According to X-ray diffraction data, all isopropoxide complexes **(L1)_2_Ti(OiPr)_2_**, **(L1)_2_Zr(OiPr)_2_**, and **(L2)_2_Zr(OiPr)_2_** have similar structure and contain two ligand molecules and two isopropoxide groups. Metal atoms in these complexes have a distorted octahedral environment by the four O atoms and two N atoms of ligands. The oxygen atoms of two phenoxy imine ligands are situated in the trans-position and two oxygen atoms of isopropoxide group are in the cis-position. Dichloride complex **(L2)_2_ZrCl_2_** also has octahedral structure with similar orientation of phenoxy imine ligands. Selected bond and angles are presented in Appendix A.

The catalytic activity of phenoxyimine complexes of titanium and zirconium was studied in ethylene polymerization. The complexes were activated using MMAO-12 (modified methylalumoxane), tetrakis-perfluorophenyl borate in combination with *i*-Bu_3_Al, and mixtures of Et_2_AlCl or Et_3_Al_2_Cl_3_ with di-*n*-butylmagnesium in a molar ratio of Al/Mg = 3:1. The most interesting results are shown in Table 1. Comparative characteristics of the effectiveness of various activators are shown in Figure 3.

MAO, a traditional co-catalyst for FI complexes, quite effectively activates titanium and zirconium dichloride complexes (experiments 5,13, 14, 21, 25, Table 1). The zirconium complex with a non-fluorinated ligand-**(L1)_2_ZrCl_2_** showed the highest activity (3200 kg of PE⸱mol^−1^ h^−1^ atm^−1^). Polymers obtained using MAO are characterized by the maximum values of molecular weights in this series-4–6 · 10^6^ Da. Using the **(L2)_2_TiCl_2_** complex as an example, the effect of the Al_MAO_/Ti ratio on the productivity of the catalytic system and the properties of the formed polymer was studied. It was found that with an increase in this ratio from 1:500 to 1:1000, the productivity slightly decreases, and the molecular weight of the polymer increases from 4.1 to 4.6 × 10^6^ (experiments 13–14). This observation is consistent with the data [38], the authors of which believe that an increase in Al/Ti ratio leads to not only an increase in the number of active centers but also increases in the polarity of the medium, which leads to better ion separation and stabilization of the ion-pair. We also do not rule out that an increase in the Al/Ti ratio promotes the reduction of Ti(IV) to Ti(III).

The Ti and Zr alkoxide complexes in the presence of MAO either showed no activity at all, or it was very insignificant.

The cocatalyst (^i^Bu)_3_Al/CPh_3_^+^B(C_6_F_5_)_4_^−^ turned out to be ineffective on the whole. Only for the complex **(L1)_2_ZrCl_2_**, an activity of 900 kg of PE⸱mol^−1^ h^−1^ atm^−1^ was recorded (the polymer has a molecular weight of about 5 million Da.

Al/Mg activators of the composition {Et_n_AlCl_3-n_ + Bu_2_Mg} with titanium phenoxy-imine complexes form highly active catalytic systems, often superior in productivity to systems with MAO. It is important that these cocatalysts effectively activate alkoxide titanium complexes. The productivity of such catalytic systems often is comparable to dichloride analogues.

For the **(L1)_2_TiCl_2_**/Et_2_AlCl+Bu_2_Mg catalytic system, the influence of the polymerization temperature on the activity and on the molecular weights of the resulting polyethylene was studied (Figure 4). It has been established that this catalytic system exhibits a sufficiently high thermal stability: with an increase in the polymerization temperature from 30 to 50 °C, the activity increases by more than 1.5 times. However, a further increase in temperature to 80 °C leads to its deactivation. With an increase in the synthesis temperature, the processes of polymer chain termination are accelerated, which is reflected in a significant decrease in the molecular weights of the polymer. Thus, the process temperature allows you to control the molecular weight of the resulting polymers.

However, the Al/Mg cocatalysts are unable to activate zirconium phenoxy-imine complexes. We assume that the ability of {Et_n_AlCl_3-n_ + Bu_2_Mg} mixtures to effectively activate titanium dichloride and dialkoxide complexes is associated with the reduction of Ti(IV) to Ti(III). Based on this assumption, the inertness of zirconium complexes with the same Al/Mg cocatalysts can be explained by significant differences in the values of the electrochemical potentials of titanium and zirconium. For example, for metallocenes Cp_2_MCl_2_: −E_1\2_ = 1.4 (Ti), 1.6–2.3 (Zr) and 2.7 (Hf) [39,40,41] and for titanium and zirconium complexes with an oxopyran ligand: −E_1\2_ = 0.48 (Ti), 1.3 (Zr) [42]. However, in the available literature, we were unable to find arguments in favor of such a version; on the contrary, the reduction of a group 4 metal is considered as one of the deactivation pathways for phenoxyimine catalysts. In any case, this version needs additional experimental confirmation.

The processes of activation of Ti and Zr complexes, even with the same ligand environment, have their own specifics. For example, the paper [14] reports that Et_2_AlCl is capable of activating the FI pre-catalyst bis[N-(3-*tert*-butylsalicylidene)-2,3,4,5,6-pentafluoroanilinato]titanium(IV) dichloride, while Et_3_Al and i-Bu_3_Al did not show activating ability. For the zirconium precatalyst-[N-(3-*tert*-butylsalicylidene)anilinato]zirconium(IV) dichloride-the pattern is reversed: only the alkylaluminum with three linear alkyls can activate it (with productivity in some cases comparable to MAO; Al/Zr = 4000–20000), while Bu_3_Al or Et_2_AlCl do not have activating properties at all [12].

### 3.1. Morphology and Particle Size of UHMWPE Reactor Powders

Morphology, particle size and their uniformity, along with molecular weight and MWD, are important characteristics of UHMWPE nascent reactor powder which determine the possibility of its processing into high-modulus and high-strength oriented materials. To examine the morphologies of these powders, SEM technique was used (Figure 5 and Appendix A). At low magnification, the polymer particles have the irregular shape and porous structure, which determines the low bulk density (0.04–0.09 g/cm^3^) of the obtained samples. For various grades of UHMWPE produced on an industrial scale, this parameter has significantly higher values. For example, a polymer GUR 2122 (Ticona, Germany) has a bulk density of 0.20–0.25 g/cm^3^ and this value is considered low [43]. At high magnification, the morphology of polymer powders obtained with catalytic systems containing MAO predominantly revealing a “wave-like” shape, which is usually a feature of relatively ductile material capable of large deformation [44]. On the contrary, polymer powders obtained with the use of Al/Mg activators are characterized by a broccoli-like morphology (Figure 5, bottom).

The method of the laser diffraction in a suspension of UHMWPE powders has been used to estimate particle size and uniformity. The size distributions were reported by the cumulative volume diameter at 10%, 50% and 90% (Figure 6). As can be seen from the results, the powders under study are characterized by a rather large particle size compared to commercially available UHMWPE samples. For example, polymer-grade GUR 2122 has the average particle size of 125 µm [45].

The smallest values of the average particle size (59–81 µm) are possessed by polymers obtained on titanium dialkoxide phenoxyimine complexes activated by {Et_2_AlCl + Bu_2_Mg}. It is obvious that the polymer particles, which have an irregular shape and porous structure, “caking” during storage, forming larger agglomerates. This is especially true for polymers obtained using MAO.

### 3.2. Mechanical Properties of Synthesized UHMWPE

The processing of UHMWPE reactor powders into high-modulus oriented films was carried out by preparing monolithic samples under pressure and shear deformation at temperatures lower than the main melting peak with subsequent uniaxial drawing [8]. The mechanical properties of oriented film tapes are given in Table 2 and Figure 7.

Some samples of UHMWPE reactor powders turned out to be unsuitable for the solid-phase processing. As can be seen from the presented data, oriented films from UHMWPE obtained on catalytic systems containing MAO are characterized by the best mechanical characteristics in this series. It is known that one of the important parameters of the UHMWPE reactor powders, which determine the high tensile strength of the disentangled tapes, are the high molar mass and narrow MWD [7]. Probably, the best mechanical characteristics of the samples obtained in experiments 3 and 4 (Table 1 and Table 2) are due to the ability of the FI/MAO systems to produce polymers with a narrow MWD. The authors [7], using the melt rheological method, showed that UHMWPE obtained on such systems is not monomodal (the classical method for determining MWD-GPC is unacceptable for UHMWPE). However, the use of Al/Mg activators leads to the formation of a set of inhomogeneous catalytically active sites, and, as a consequence, to an even more significant MWD broadening [46,47,48].

Oriented films from UHMWPE powders obtained using Al/Mg activators show fairly good mechanical characteristics (tensile strength up to 2.3 GPa; tensile modulus up to 147 GPa). For comparison, the modulus value for commercially available gel-spun UHMWPE fiber, produced by gel-spinning process, is 113 GPa [49]. In this series of UHMWPE samples, in general, the following trend is traced: the best mechanical characteristics were shown by polymers obtained on a polyfluorinated precatalyst (compare experiments 1 vs. 11 and 4 vs. 12). We believe that this is a manifestation of the “fluorine” effect-the ability of the fluorine atoms of the precatalyst to form weak agnostic interactions with the hydrogen atoms of the growing polymer chain [50,51,52], which prevents both β-hydrogen elimination and β-hydrogen transfer to the monomer, and in particular, contribute to the narrowing of the MWD.

After achieving good results with the polymerization of ethylene, we investigated the activity of titanium FI complexes in the presence of Al/Mg activators in the copolymerization of ethylene with octene-1 and ternary copolymerization ethylene/propylene/5-vinyl-2-norbornene. The obtained copolymers were characterized by DSC, GPC, ^1^H- and ^13^C-NMR. The results are summarized in Table 3 and Table 4.

Catalytic systems **FI**/Et_n_AlCl_3-n_ + Bu_2_Mg produce high ultrahigh molecular weight ethylene/1-octene copolymers with productivity up to 2.7 tons mol(Ti)^−1^·h^−1^ atm^−1^. In contrast to the considered catalytic systems, FI/MAO do not produce ultrahigh molecular weight copolymers. For example, the authors [52] reported living polymerization of Ethylene/1-Octene with bis[N-(3-methylsalicylidene)-2,3,4,5,6-pentafluoroanilinato]TiCl_2_/dried MAO; comonomer content reached 32.7 mol% depending the 1-octene feeding ratio, but the M_w_ values did not exceed 4.9 × 10^5^ Da.

The inclusion of the comonomer is in the range of 2.3–3.8 mol%. The MWD of copolymers is quite wide, which is typical for polymers obtained using Al/Mg activators; however, catalytic system **(L1)_2_Ti(OiPr)_2_**/Et_3_Al_2_Cl_3_+Bu_2_Mg produces a copolymer with the narrowest MWD-2.3. The same system is the best in terms of productivity and the degree of comonomer inclusion. In this series of experiments, no advantages of complexes with polyfluorinated ligands were revealed.

Such copolymers are of undoubted practical interest, since the introduction of small amounts of long-chain branches into the UHMWPE macromolecule can facilitate its processing. (UHMWPE is a perspective engineering polymer with a unique set of properties and applications. However, the high molecular weight and the presence of a large number of entanglements between chains cause a very high melt viscosity, which greatly complicates the processing of this polymer by conventional methods). For the synthesis of such E/O copolymers, various catalytic systems are used, for example, (F-salalen)TiCl_2_/MAO [53], hydrocarbyl Zr and Hf complexes supported by α-diimine ligands with the camphyl linker/[Ph_3_C][B(C_6_F_5_)_4_] [54].

Our systems are significantly superior to the mentioned analogs in terms of activity and are not inferior to them in terms of molecular weights and comonomer content.

Ethylene-propylene-diene terpolymers (EPDM) are one of the popular synthetic rubbers. The Ti-FI complexes in the presence of Et_3_Al_2_Cl_2_/Bu_2_Mg catalyzed the ternary copolymerization of ethylene, propylene, and 5-vinyl-2-norbornene (Table 4). Tercopolymers with a content of propylene units up to 58 mol% and 5-VBN up to 5 mol% were synthesized; however, the productivity of systems with FI complexes is significantly inferior to titanium complexes with 2-hydroxymethylphenol derivatives with the same Al/Mg activators [55].

The synthesized copolymers are characterized by relatively M_w_ weights (from 3.9 × 10^5^ to 5.8 × 10^5^ Da), broad molecular weight distributions and low degrees of crystallinity. It is interesting that the copolymers obtained on alkoxide complexes (entries 2 and 4, Table 4, Appendix A) are characterized by a minimum MWD value.

## 4. Conclusions

A series of Ti(IV) and Zr(IV) dichloride and dialkoxide complexes with phenoxyimine ligands (including polyfluorinated ones) have been synthesized. The structure of the four coordination compounds was determined by X-ray diffraction.

The effects of the nature of the activator (MAO-12, (^i^Bu)_3_Al/CPh_3_^+^B(C_6_F_5_)_4_^−^ and combinations of alkylaluminum chlorides and dibutylmagnesium) on the productivity of catalytic systems in the polymerization of ethylene and its copolymerization with 1-octene and the ternary copolymerization of ethylene, propylene and 5-vinyl-2-norbornene, as well as on the properties of the resulting polymers were studied.

It is found that

Al/Mg co-catalysts (mixtures of alkylaluminum chlorides with dibutylmagnesium) are effective activators of Ti(IV) phenoxy-imine complexes regardless of the nature of the σ-ligand (Cl or OiPr);The productivity of such systems, as a rule, exceeds similar systems containing MAO;MAO and (^i^Bu)_3_Al/CPh_3_^+^B(C_6_F_5_)_4_^−^ practically does not activate dialkoxide complexes of titanium and zirconium, while Al/Mg co-catalysts form quite active Ti-containing systems;Unfortunately, Al/Mg co-catalysts are not able to activate zirconium-containing systems;The UHMWPE nascent reactor powders, obtained on titanium phenoxyimine complexes, can be processed by the solventless solid-phase method into high-strength, high-modulus oriented films. It should be noted that the mechanical characteristics of oriented materials obtained using Al/Mg activators are still inferior to analogs obtained using MAO;Polymers obtained on zirconium-containing systems turned out to be unsuitable for solid-phase processing;**(FI)_2_TiX_2_**/Et_n_AlCl_3-n_ + Bu_2_Mg catalytic systems produce ultra-high molecular weight ethylene-1-octene copolymers and ethylene/propylene/5-vinyl-2-norbornene terpolymers. Polymers obtained on dialkoxide complexes (X = O-iPr) are characterized by a narrow MWD (2–2.5), while for polymers obtained on dichloride complexes, this value is much higher (5.2–10.2).

Thus, the mixtures {Et_n_AlCl_n-3_ + Bu_2_Mg} are quite a worthy replacement for expensive activators-alumoxanes and perfluorophenylborates. Work on optimizing the compositions of Al/Mg activators capable of activating not only titanium but also zirconium complexes are ongoing.

## Data Availability

The data presented in this study are available from the corresponding author upon request.

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
