# Peer review of "Effect of Activator and Outgoing Ligand Nature on the Catalytic Behavior of Bis(phenoxy-imine) Ti(IV) Complexes in the Polymerization of Ethylene and Its Copolymerization with Higher Olefins"

_polymers, 2022, doi:10.3390/polym14204397_

Round 1
Reviewer 1 Report
In this manuscript, Gagieva and coworker report on the synthesis and characterization of phenoxy-imine (FI) complexes of Ti and Zr having either -Cl- or -OiPr ancillary ligands and on their use as pre-catalysts in the homo/copolymerization of ethylene. The authors studied the effect of different activators on the catalytic performance of the complexes along with the properties of the produced materials.
I find the work complete, well written and nicely presented; I am hence favorable to its publication in Polymers as is.
Editing issues to be assessed:
At the very beginning of the abstract, please state what “FI” stands for.
In the experimental part, indicate how the crystals suitable for Xray analyses were obtained
First line of section 3.1: is there some text missing?
The drawing of the complexes in scheme 1 could be presented in a more pleasant way
In figure 3 and 7, adding a second type of labelling (along with the color-coding) would help
Author Response
October 11, 2022
Manuscript Number: polymers-1956520
Dear Dr. Zoe Ou,
Thank you for your kind considering of our manuscript!
We thank Reviewers for providing constructive comments and help in improving the content of this paper. According to their comments, we made the following revisions (all changes in the text are highlighted in red):
Reviewer #1:
- " At the very beginning of the abstract, please state what “FI” stands for”.
Recommended explanation has been added.
- “In the experimental part, indicate how the crystals suitable for X-ray analyses were obtained
The conditions for obtaining single crystals suitable for X-ray were added to the experimental part: «The single crystal for XRD studies was obtained by recrystallization of the resulting crystalline powder from a mixture of toluene and hexane (1:2)».
- First line of section 3.1: is there some text missing?
We have rechecked the first line of section 3.1- (Results and discussion) does not contain errors and the phrase is finished. «Phenoxyimine ligands L1 and L2 containing bulky tert-butyl groups in the ortho- and para-positions to the phenolic hydroxyl were used as objects of study in this work. These substituents protect the metal center, stabilize the catalytically active sites, and ensure good solubility of the complex».
The drawing of the complexes in scheme 1 could be presented in a more pleasant way
The drawing of the complexes in scheme 1 has been changed.
In figure 3 and 7, adding a second type of labelling (along with the color-coding) would help.
In figure 7 a second type of labelling (along with the color-coding) have been added
On behalf of all authors,
Svetlana Gagieva,
Leading Researcher
Moscow State University,
119991 Moscow, Russia
Fax: +7(495)9393316
E-mail: sgagieva@yandex.ru

Reviewer 2 Report
The goal of the work is the synthesis of series of bis(phenoxy-imine) titanium(IV) and zirconium(IV) complexes for ethylene homo- and co-polymerization with higher alpha-olefins. An elegant activation of Ti-Fi complexes with Al/Mg activators for production of UHMWPE and copolymers is demonstrated. Mechanical properties of synthesized UHMWPE were tested and it was shown that several samples of UHMWPE are suitable for the solid-phase processing. This high quality manuscript is actual. I would like to recommend this manuscript to be accepted for publication in Polymer after minor corrections.
Please, find my questions and comments below:
1) Technical correction: in Experimental section, item 3.1: please, check a formatting of degree of Celsius (now degree of Celsius is in the middle position: to my mind it should be “90 °C” instead of “90 â—¦C”)
2) The MWD plots of polymers, presented in tales 3 and 4, are extremely desirable. From tables 3 an4 it is clear that MWD of some copolymers is much broader than theoretical Mw/Mn=2, typical for most probably Flory distribution. MWD plots can demonstrate the effect of complexes structure and type of activator on MWD broadening (simple broadening or even becoming of bimodal).
Author Response
October 11, 2022
Manuscript Number: polymers-1956520
Dear Dr. Zoe Ou,
Thank you for your kind considering of our manuscript!
We thank Reviewers for providing constructive comments and help in improving the content of this paper. According to their comments, we made the following revisions (all changes in the text are highlighted in red):
Reviewer #2:
- Technical correction: in Experimental section, item 3.1: please, check a formatting of degree of Celsius (now degree of Celsius is in the middle position: to my mind it should be “90 °C” instead of “90 â—¦C”)
We corrected it.
- The MWD plots of polymers, presented in tales 3 and 4, are extremely desirable. From tables 3 an4 it is clear that MWD of some copolymers is much broader than theoretical Mw/Mn=2, typical for most probably Flory distribution. MWD plots can demonstrate the effect of complexes structure and type of activator on MWD broadening (simple broadening or even becoming of bimodal).
The main objective of this work was to develop a new post-metallocene catalytic system, effective in the copolymerization of ethylene with 1-octene. This task has been completed and its main results are presented in the submitted manuscript. However, in the process of optimizing these systems, it was found that almost all of the obtained copolymers have very high molecular weights (Mw ~ 3 106) and a relatively wide MWD (Mw/Mn ratios from 2.25 to 10) with a clearly bimodal character (the GPC curves are given in Supporting Materials, Figures S41-S44). This indicates the multicenter nature of the emerging catalytic system. In this case, a clear dependence of the degree of multicenterness on the composition and amount of the activator is traced, since a relatively narrow value of the molecular weight distribution (Mw/Mn = 2.25) was obtained for one of the copolymers. These facts are quite interesting from practical and scientific points of view, and they prompted us to continue the study and optimization of these systems and to conduct a more thorough analysis of the structure and properties of the obtained copolymers, including the study of the Flory distribution. As a result of this intention, we have excluded from this manuscript some preliminary results on the properties of polymers and, after performing additional planned experiments, we will devote a separate publication to this issue.
On behalf of all authors,
Svetlana Gagieva,
Leading Researcher
Moscow State University,
119991 Moscow, Russia
Fax: +7(495)9393316
E-mail: sgagieva@yandex.ru
